# Application of First-Order Shear Deformation Theory on Vibration Analysis of Stepped Functionally Graded Paraboloidal Shell with General Edge Constraints

**DOI:** 10.3390/ma12010069

**Published:** 2018-12-25

**Authors:** Fuzhen Pang, Haichao Li, Fengmei Jing, Yuan Du

**Affiliations:** College of Shipbuilding Engineering, Harbin Engineering University, Harbin 150001, China; pangfuzhen@hrbeu.edu.cn (F.P.); duyuan@hrbeu.edu.cn (Y.D.)

**Keywords:** stepped FG paraboloidal shell, general edge conditions, spring stiffness technique, free vibration characteristics

## Abstract

The paper introduces a semi-analytical approach to analyze free vibration characteristics of stepped functionally graded (FG) paraboloidal shell with general edge conditions. The analytical model is established based on multi-segment partitioning strategy and first-order shear deformation theory. The displacement components along axial direction are represented by Jacobi polynomials, and the Fourier series are utilized to express displacement components in circumferential direction. Based on penalty method about spring stiffness technique, the general edge conditions of doubly curved paraboloidal shell can be easily simulated. The solutions about doubly curved paraboloidal shell were solved by approach of Rayleigh–Ritz. Convergence study about boundary parameters, Jacobi parameters et al. are carried out, respectively. The comparison with published literatures, FEM and experiment results show that the present method has good convergence ability and excellent accuracy.

## 1. Introduction

The stepped FG paraboloidal shells are very useful in the engineering. The vibration problems of the structures have always been the concern of the research: Fantuzzi et al. [1] investigated free vibration behavior of FG cylindrical and spherical shells. On the base of FSDT, Tornabene and Reddy [2] used the GDQ approach to investigate the vibration behavior of FGM shells and panels. Based on higher-order finite element method, Pradyumna and Bandyopadhyay [3] studied the vibration behavior of FG structures. Jouneghani et al. [4] also investigated the characteristics of FG doubly curved shells. Chen et al. [5] obtained the vibration characteristics of FG sandwich structure based on shear deformation theory. Wang et al. [6,7,8,9] investigated the approach of Improved Fourier to study vibration phenomenon of various structures. Tornabene et al. [10,11,12] used the GDQ method to research four parameter FG composite structures. Fazzolari and Carrera [13] solved the vibration issues of FG structures based on Ritz minimum energy approach. Kar and Panda [14] studied vibration characteristics of FG spherical shell by FEM. Tornabene [15] focused on the dynamic behavior of FG structures. Zghal [16] investigated the vibration characteristics of FG shells. Kulikov et al. [17] dealt with a recently developed approach to analyze free vibration behavior of FG plates by the formulations of sampling surfaces. Kapuria et al. [18] developed a four-node quadrilateral element method to analyze dynamic vibration of FGM shallow shells.

In field of FG stepped shells, Hosseini-Hashemi et al. [19] proposed an accurate solution to study vibration characteristics of stepped FG plates. Bambill et al. [20] solved vibrations behavior of axially FG beams with stepped changes in geometry. Vinyas and Kattimani [21,22] carried out the static analysis of stepped FG beam and plates with various loads. Su et al. [23] presented an effective method to study free vibration of stepped FG beams.

From literatures reviewed, we can find that many scholars applied Rayleigh Ritz method, GDQ method, Improved Fourier series method, FEM and Haar Wavelet Discretization method etc. to study vibration characteristics of FG doubly curved structures. There are no literatures put attentions on free vibration problems of stepped FG paraboloidal shell. So, it is very important to propose a unified formulation to study free vibration behaviors of stepped FG paraboloidal shell subject to general edge conditions.

## 2. Fundamental Theory

### 2.1. The Description of the Model

The model of stepped FG paraboloidal shell is described in Figure 1. *h_i_* represents the thickness of the structure. The stepped structure is obtained by the curve *c*_1_*c*_2_. The model is established on the basis of orthogonal coordinate system (*φ*, *θ*, *z*), which represent axial, circumferential and normal directions, respectively. The displacements are represented by *u*, *v* and *w*, respectively.

The doubly-curved paraboloidal shell is shown in Figure 2. The displacement components of stepped FG paraboloidal shell are represented by U, V and W. In addition, the doubly curved paraboloidal shell is divided into *H* shell segments along axial direction [24,25].

The Young’s modulus *E*, Poisson’s ratios *ν* and mass density *ρ* of two typical FG models are shown as follow [26,27,28,29,30,31,32]:(1a)$$E(z)=(E_{c}-E_{m})V_{c}+E_{m}$$
(1b)$$\rho (z)=(\rho_{c}-\rho_{m})V_{c}+\rho_{m}$$
(1c)$$\nu (z)=(\nu_{c}-\nu_{m})V_{c}+\nu_{m}$$
where *c* and *m* denote the ceramic and metallic constituents, respectively. The volume fractions *V_c_* are shown as follow [33]:(2a)$$FGM_{I}\left(a/b/c/p\right):V_{c}={\left[1-a\left(\frac{1}{2}+\frac{z}{h}\right)+b{\left(\frac{1}{2}+\frac{z}{h}\right)}^{c}\right]}^{p}$$
(2b)$$FGM_{\mathrm{II}}\left(a/b/c/p\right):V_{c}={\left[1-a\left(\frac{1}{2}-\frac{z}{h}\right)+b{\left(\frac{1}{2}-\frac{z}{h}\right)}^{c}\right]}^{p}$$
where *z* and *p* represent the thickness and power law exponent of the structure, respectively. We should note that the value of parameter *p* takes only positive values. The symbols *a*, *b* and *c* are the key parameters which affect the property of FG material largely. As the volume fraction, the total value of which should be the one. From Equations (1) and (2), we can easily get that the functionally graded material will be the isotropic material when the power law exponent equal to infinity or zero. The variations *V_c_* about various values of *a*, *b*, *c* and *p* are showed in Figure 3. In addition, we should note that the distributions of volume fraction (2a) and (2b) are mirror reflections. Thus, the Variations *V_c_* of FGM_II_ are ignored in Figure 3. The detailed descriptions of FG material are reported in Refs. [34,35,36].

### 2.2. Energy Equations of Stepped FG Paraboloidal Shell

The displacements of *i*th segment in stepped FG paraboloidal shell are shown as below:(3a)$${\bar{U}}^{i}\left(\varphi ,\theta ,z,t\right)=u^{i}\left(\varphi ,\theta ,t\right)+z\psi_{\varphi }^{i}\left(\varphi ,\theta ,t\right)$$
(3b)$${\bar{V}}^{i}\left(\varphi ,\theta ,z,t\right)=v^{i}\left(\varphi ,\theta ,t\right)+z\psi_{\theta }^{i}\left(\varphi ,\theta ,t\right)$$
(3c)$${\bar{W}}^{i}\left(\varphi ,\theta ,z,t\right)=w^{i}\left(\varphi ,\theta ,t\right)$$

The strains of stepped FG paraboloidal shell are shown as follow
(4a)$$\varepsilon_{\varphi }^{i}=\varepsilon_{\varphi }^{0,i}+z\kappa_{\varphi }^{0,i}\varepsilon_{\theta }^{i}=\varepsilon_{\theta }^{0,i}+z\kappa_{\theta }^{0,i}$$
(4b)$$\gamma_{\varphi \theta }^{i}=\gamma_{\varphi \theta }^{0,i}+z\kappa_{\varphi \theta }^{0,i}\gamma_{\varphi z}^{i}=\gamma_{\varphi z}^{0,i}\gamma_{\theta z}^{i}=\gamma_{\theta z}^{0,i}$$
where $\varepsilon_{\varphi }^{i}$, $\varepsilon_{\theta }^{i}$, $\varepsilon_{\varphi \theta }^{i}$, $\gamma_{\varphi z}^{0,i}$, $\gamma_{\theta z}^{0,i}$, $k_{\varphi }^{i}$, $k_{\theta }^{i}$ and $k_{\varphi \theta }^{i}$ are given as:(5a)$$\varepsilon_{\varphi }^{0,i}=\frac{1}{A}\frac{\partial u^{i}}{\partial \varphi }+\frac{v^{i}}{AB}\frac{\partial A}{\partial \theta }+\frac{w^{i}}{R_{\varphi }}$$
(5b)$$\varepsilon_{\theta }^{0,i}=\frac{1}{B}\frac{\partial v^{i}}{\partial \theta }+\frac{u^{i}}{AB}\frac{\partial B}{\partial \varphi }+\frac{w^{i}}{R_{\theta }}$$
(5c)$$\gamma_{\varphi \theta }^{0,i}=\frac{A}{B}\frac{\partial }{\partial \theta }\left(\frac{u^{i}}{A}\right)+\frac{B}{A}\frac{\partial }{\partial \varphi }\left(\frac{v^{i}}{B}\right)$$
(5d)$$\kappa_{\varphi }^{i}=\frac{1}{A}\frac{\partial \psi_{\varphi }^{i}}{\partial \varphi }+\frac{\psi_{\theta }^{i}}{AB}\frac{\partial A}{\partial \theta }$$
(5e)$$\kappa_{\theta }^{i}=\frac{1}{B}\frac{\partial \psi_{\theta }^{i}}{\partial \theta }+\frac{\psi_{\varphi }^{i}}{AB}\frac{\partial B}{\partial \varphi }$$
(5f)$$\kappa_{\varphi \theta }^{i}=\frac{A}{B}\frac{\partial }{\partial \theta }\left(\frac{\psi_{\varphi }^{i}}{A}\right)+\frac{B}{A}\frac{\partial }{\partial \varphi }\left(\frac{\psi_{\theta }^{i}}{B}\right)$$
(5g)$$\gamma_{\varphi z}^{0,i}=\frac{1}{A}\frac{\partial w^{i}}{\partial \varphi }-\frac{u^{i}}{R_{\varphi }}+\psi_{\varphi }^{i}$$
(5h)$$\gamma_{\theta z}^{0,i}=\frac{1}{B}\frac{\partial w^{i}}{\partial \theta }-\frac{v^{i}}{R_{\theta }}+\psi_{\theta }^{i}$$

For doubly curved paraboloidal shell, the symbols *A* and *B* are shown as below [37,38]: (6)$$A=R_{\varphi },B=R_{\theta }\sin\varphi$$

Based on Hooke’s law, the stresses corresponding to strains can be expressed as:(7)$$\left\{\begin{array}{c} \sigma_{\varphi }^{i}\\ \sigma_{\theta }^{i}\\ \tau_{\varphi \theta }^{i}\\ \tau_{\varphi z}^{i}\\ \tau_{\theta z}^{i} \end{array}\right\}=\left[\begin{array}{ccccc} Q_{11}\left(z\right) & Q_{12}\left(z\right) & 0 & 0 & 0\\ Q_{12}\left(z\right) & Q_{11}\left(z\right) & 0 & 0 & 0\\ 0 & 0 & Q_{66}\left(z\right) & 0 & 0\\ 0 & 0 & 0 & Q_{66}\left(z\right) & 0\\ 0 & 0 & 0 & 0 & Q_{66}\left(z\right) \end{array}\right]\left\{\begin{array}{c} \varepsilon_{\varphi }^{i}\\ \varepsilon_{\theta }^{i}\\ \gamma_{\varphi \theta }^{i}\\ \gamma_{\varphi z}^{i}\\ \gamma_{\theta z}^{i} \end{array}\right\}$$
where $\sigma_{\varphi }^{i}$ and $\sigma_{\theta }^{i}$ are normal stresses; $\tau_{\varphi \theta }^{i}$, $\tau_{\varphi z}^{i}$ and $\tau_{\theta z}^{i}$ are shear stresses. The *Q_ij_*(*z*) are defined as follows:(8)$$Q_{11}(z)=\frac{E(z)}{1-\nu^{2}(z)},Q_{12}(z)=\frac{\nu (z)E(z)}{1-\nu^{2}(z)},Q_{66}(z)=\frac{E(z)}{2[1+\nu (z)]}$$

The force and moment resultants can be obtained as follow:(9a)$$\left\{\begin{array}{c} N_{\varphi }^{i}\\ N_{\theta }^{i}\\ N_{\varphi \theta }^{i} \end{array}\right\}=\left[\begin{array}{ccc} A_{11} & A_{12} & 0\\ A_{12} & A_{22} & 0\\ 0 & 0 & A_{66} \end{array}\right]\left\{\begin{array}{c} \varepsilon_{\varphi }^{0,i}\\ \varepsilon_{\theta }^{0,i}\\ \gamma_{\varphi \theta }^{0,i} \end{array}\right\}+\left[\begin{array}{ccc} B_{11} & B_{12} & 0\\ B_{12} & B_{22} & 0\\ 0 & 0 & B_{66} \end{array}\right]\left\{\begin{array}{c} \varepsilon_{\varphi }^{0,i}\\ \varepsilon_{\theta }^{0,i}\\ \gamma_{\varphi \theta }^{0,i} \end{array}\right\}$$
(9b)$$\left\{\begin{array}{c} M_{\varphi }^{i}\\ M_{\theta }^{i}\\ M_{\varphi \theta }^{i} \end{array}\right\}=\left[\begin{array}{ccc} B_{11} & B_{12} & 0\\ B_{12} & B_{22} & 0\\ 0 & 0 & B_{66} \end{array}\right]\left\{\begin{array}{c} \varepsilon_{\varphi }^{0,i}\\ \varepsilon_{\theta }^{0,i}\\ \gamma_{\varphi \theta }^{0,i} \end{array}\right\}+\left[\begin{array}{ccc} D_{11} & D_{12} & 0\\ D_{12} & D_{22} & 0\\ 0 & 0 & D_{66} \end{array}\right]\left\{\begin{array}{c} \kappa_{\varphi }^{i}\\ \kappa_{\theta }^{i}\\ \kappa_{\varphi \theta }^{i} \end{array}\right\}$$
(9c)$$\left\{\begin{array}{c} Q_{\varphi }^{i}\\ Q_{\theta }^{i} \end{array}\right\}=\bar{\kappa }\left[\begin{array}{cc} A_{66} & 0\\ 0 & A_{66} \end{array}\right]\left[\begin{array}{c} \gamma_{\varphi z}^{0,i}\\ \gamma_{\theta z}^{0,i} \end{array}\right]$$
where $\bar{\kappa }$ is shear correction factor. $A_{ij}$, $B_{ij}$ and $D_{ij}$ are obtained by following integral:(10)$$(A_{ij},B_{ij},D_{ij})=\int_{-h/2}^{h/2}Q_{ij}(z)(1,z,z^{2})dz$$

The strain energy of the select segment can be expressed from Equation (11) as shown:(11)$$U^{i}=\frac{1}{2}{∭}_{V}\left(\begin{array}{l} N_{\varphi }^{i}\varepsilon_{\varphi }^{0,i}+N_{\theta }^{i}\varepsilon_{\theta }^{0,i}+N_{\varphi \theta }^{i}\gamma_{\varphi \theta }^{0,i}+M_{\varphi }^{i}k_{\varphi }^{i}+\\ M_{\theta }^{i}k_{\theta }^{i}+M_{\varphi \theta }^{i}k_{\varphi \theta }^{i}+Q_{\varphi }^{i}\gamma_{\varphi z}^{0,i}+Q_{\theta }^{i}\gamma_{\theta z}^{0,i} \end{array}\right)ABd\varphi d\theta dz$$

To save the space of this paper, the Equation (11) can be expressed as $U^{i}=U_{S}^{i}+U_{B}^{i}+U_{BC}^{i}$. The detailed description of $U_{S}^{i}$, $U_{B}^{i}$ and $U_{BC}^{i}$ are shown in Appendix A.

The maximum kinetic energy of the select segment can be obtained from Equation (12) as shown:(12)$$\begin{array}{l} T^{i}=\frac{1}{2}{∭}_{V}\rho \left(z\right)\left[{\left({\dot{\bar{U}}}^{i}\right)}^{2}+{\left({\dot{\bar{V}}}^{i}\right)}^{2}+{\left({\dot{\bar{W}}}^{i}\right)}^{2}\right]\left(1+\frac{z}{R_{\varphi }}\right)\left(1+\frac{z}{R_{\theta }}\right)ABd\varphi d\theta dz=\left[\;\;\right]\\ =\frac{1}{2}\int_{\varphi_{0}}^{\varphi_{1}}\int_{0}^{2\pi }\left\{I_{0}\left[{\left({\dot{u}}^{i}\right)}^{2}+{\left({\dot{v}}^{i}\right)}^{2}+{\left({\dot{w}}^{i}\right)}^{2}\right]+2I_{1}\left({\dot{u}}^{i}{\dot{\psi }}_{\varphi }^{i}+{\dot{v}}^{i}{\dot{\psi }}_{\theta }^{i}\right)+I_{2}\left[{\left({\dot{\psi }}_{\varphi }^{i}\right)}^{2}+{\left({\dot{\psi }}_{\theta }^{i}\right)}^{2}\right]\right\}ABd\varphi d\theta \end{array}$$
where the dot denotes the differentiation about time, whereas three integrals are defined as follows:(13)$$\left(I_{0},I_{1},I_{2}\right)=\int_{-h/2}^{h/2}\rho \left(z\right)\left(1+\frac{z}{R_{\varphi }}\right)\left(1+\frac{z}{R_{\theta }}\right)\left(1,\;z,\;z^{2}\right)dz$$

The energy in two sides of boundary springs can be expressed as:(14)$$\begin{array}{l} U_{b}=\frac{1}{2}{\int_{0}^{2\pi }\int_{-h/2}^{h/2}\left\{k_{u,0}u_{}^{2}+k_{v,0}v_{}^{2}+k_{w,0}w_{}^{2}+k_{\varphi ,0}\psi_{\varphi }^{2}+k_{\theta ,0}\psi_{\theta }^{2}\right\}}_{\varphi =\varphi_{r,0}}Bd\theta dz\\ +\frac{1}{2}{\int_{0}^{2\pi }\int_{-h/2}^{h/2}\left\{k_{u,1}u_{}^{2}+k_{v,1}v_{}^{2}+k_{w,1}w_{}^{2}+k_{\varphi ,1}\psi_{\varphi }^{2}+k_{\theta ,1}\psi_{\theta }^{2}\right\}}_{\varphi =\varphi_{r,1}}Bd\theta dz \end{array}$$
where $k_{t,0}\;(t=u,v,w,\varphi ,\theta )$ and $k_{t,1}$ denote the value of springs at two sides.

The energy in connective springs of two neighbor segments is expressed as:(15)$$U_{s}^{i}=\frac{1}{2}{\int_{0}^{2\pi }\int_{-h/2}^{h/2}\left\{\begin{array}{l} k_{u}{\left(u_{}^{i}-u_{}^{i+1}\right)}^{2}+k_{v}{\left(v_{}^{i}-v_{}^{i+1}\right)}^{2}+k_{w}{\left(w_{}^{i}-w_{}^{i+1}\right)}^{2}\\ +k_{\varphi }{\left(\psi_{\varphi }^{i}-\psi_{\varphi }^{i+1}\right)}^{2}+k_{\theta }{\left(\psi_{\theta }^{i}-\psi_{\theta }^{i+1}\right)}^{2} \end{array}\right\}}_{i,i+1}Bd\theta dz$$

The total energy of the constraint conditions can be expressed as:(16)$$U_{BC}=U_{b}+\sum_{i=1}^{H-1}U_{s}^{i}$$

### 2.3. Displacement Functions and Solution

Proper selection of the admissible displacement function is a critical factor for the accuracy of final solution [39,40,41,42,43]. As displayed in literatures [44,45], classical Jacobi polynomials are valued in range of ϕ∈[−1,1]. Typical Jacobi polynomials $P_{i}^{\left(\alpha ,\beta \right)}(\phi )$ of degree i are shown as below in present method.
(17a)$$P_{0}^{\left(\alpha ,\beta \right)}(\phi )=1$$
(17b)$$P_{1}^{\left(\alpha ,\beta \right)}(\phi )=\frac{\alpha +\beta +2}{2}\phi -\frac{\alpha -\beta }{2}$$
(17c)$$\begin{array}{l} P_{i}^{\left(\alpha ,\beta \right)}(\phi )=\frac{(\alpha +\beta +2i-1)\left\{\alpha^{2}-\beta^{2}+\phi (\alpha +\beta +2i)(\alpha +\beta +2i-2)\right\}}{2i(\alpha +\beta +i)(\alpha +\beta +2i-2)}P_{i-1}^{\left(\alpha ,\beta \right)}(\phi )\\ -\frac{\left(\alpha +i-1)(\beta +i-1)(\alpha +\beta +2i\right)}{i(\alpha +\beta +i)(\alpha +\beta +2i-2)}P_{i-2}^{\left(\alpha ,\beta \right)}(\phi ) \end{array}$$
where α,β>−1 and i=2,3,…

Thus, the displacement functions of shell segments can be written in form of Equation (18) as shown:(18a)$$u=\sum_{m=0}^{M}U_{m}P_{m}^{\left(\alpha ,\beta \right)}(\phi )\cos(n\theta )e^{i\omega t}$$
(18b)$$v=\sum_{m=0}^{M}V_{m}P_{m}^{\left(\alpha ,\beta \right)}(\phi )\sin(n\theta )e^{i\omega t}$$
(18c)$$w=\sum_{m=0}^{M}W_{m}P_{m}^{\left(\alpha ,\beta \right)}(\phi )\cos(n\theta )e^{i\omega t}$$
(18d)$$\psi_{\varphi }=\sum_{m=0}^{M}\psi_{\varphi m}P_{m}^{\left(\alpha ,\beta \right)}(\phi )\cos(n\theta )e^{i\omega t}$$
(18e)$$\psi_{\theta }=\sum_{m=0}^{M}\psi_{\theta m}P_{m}^{\left(\alpha ,\beta \right)}(\phi )\cos(n\theta )e^{i\omega t}$$
where $U_{m}$, $V_{m}$, $W_{m}$, $\psi_{\varphi m}$ and $\psi_{\theta m}$ are unknown coefficients. *n* and *m* denote the semi wave number in axial and circumferential direction, respectively. *M* is highest degrees of semi wave number *m*. The total Lagrangian energy functions *L* can be obtained as it is shown in Equation (19):(19)$$L=\sum_{i=1}^{H}\left(T_{}^{i}-U_{}^{i}\right)-U_{BC}$$

The total Lagrangian energy function *L* is shown in Equation (20):(20)$$\frac{\partial L}{\partial \vartheta }=0\;\;\;\vartheta =U_{m},V_{m},W_{m},\psi_{\varphi m},\psi_{\theta m}$$

Substituting Equations (11), (12), (16), (18), (19) into Equation (20), then Equation (21) can be obtained as:(21)$$\left(K-\omega^{2}M\right)Q=0$$
where **K** and **M** denote stiffness and mass matrixes, respectively. **Q** is unknown coefficient matrix. 

## 3. Analysis of Examples

The general boundary conditions are denoted by the abbreviations. Thus the abbreviations F, C, SD, SS and Ei respectively represent free, clamped, shear diaphragm, shear support and elastic boundary conditions. The material properties are chosen as: $E_{m}=70\;\mathrm{GPa}$, $E_{c}=168\;\mathrm{GPa}$, $\rho_{c}=5700\;\mathrm{kg}/m^{3}$, $\rho_{m}=2707\;\mathrm{kg}/m^{3}$, $\nu_{m}=\nu_{c}=0.3$, M=8, α=0, β=−0.5, H=5. The geometrical dimensions are chosen as follows: *R*_0_ = 0.2 m, *R*_1_ = 1 m, *L_p_* = 1 m, $h_{1}:h_{2}:h_{3}:h_{4}:h_{5}=0.04:0.045:0.05:0.055:0.06$. The results of this paper are handle by: $\Omega =\omega R_{1}\sqrt{\rho_{c}/E_{c}}$.

### 3.1. Convergence Analysis

Figure 4 shows the frequency parameter of stepped FGM_I_ (*a* = 1; *b* = −0.5; *c* = 2; *p* = 2) doubly curved paraboloidal shell with different boundary parameters. We can get that the spring stiffness values in range of 10–10^10^
*E_c_* can converge to stable, regardless of the kinds of spring. In other words, for clamped boundary condition, the spring stiffness can be assigned within the range of 10–10^10^
*E_c_*. Based on the boundary parameters analysis, the general edge constraints are be provided as shown Table 1.

The relative percentage errors of stepped FGM_I_ (*a* = 1; *b* = −0.5; *c* = 2; *p* = 2) paraboloidal shell with various Jacobi parameters are presented in Figure 5. The results of *α* = *β* = 0 are selected as the reference values. We can easily conclude from Figure 5 that different Jacobi parameters will lead to almost the same results when *n* is a fixed value. The maximum relative error is less than 8 × 10^−8^. Thus, we can conclude that displacement functions consisting with Jacobi polynomial and Fourier series are perfectly appropriate. The most advantages of proposed method are the unified Jacobi polynomials, which make the displacement functions easier to select in contrast with other approaches. Figure 6 exhibits the results of stepped FG paraboloidal shell about truncation. We can get that the convergent results can be guaranteed when *M* is higher than 5. *M* is defined as the value of eight in this paper.

Table 2 exhibits the frequency parameter Ω of FGM_I_ (*a* = 1; *b* = 0; *c*; *p*) about the value of *H*, and the verification model is a spherical shell. The results are compared with those in literature [46]. From Table 2, we can conclude that the results will converge quickly as the value of *H* increase. We can also conclude that very high value of *M* is unnecessary. In addition, it can be obtained from Table 2 that the present method is strongly agreed with reference data.

### 3.2. Free vibration Behavior of Stepped FG Paraboloidal Shell

Table 3 shows the precision of the approach in solving free vibration behavior of stepped FG paraboloidal shell with clamed boundary condition, and all the FEM commercial program ABAQUS (S4R model) results have converged to stable when the element size is chosen as 0.03 m. In addition, it should be note that the homogeneous elements not graded elements [47] were used in this paper. From the comparison study, we can conclude that the present method is capable to analyze the vibration behaviors of stepped doubly curved paraboloidal shell with general boundary conditions.

To further prove the effectiveness of this method, the experiment test focused on free vibration of cylindrical shell was carried out. It should be note that the cylindrical shell is isotropic. The material properties and geometrical parameters are chosen as: *E* = 210 GPa, *ρ* = 7850 kg/m^3^, *ν* = 0.3, *R* = 0.06 m, *L* = 0.3 m, *h* = 0.005 m. The boundary condition is free for isotropic cylindrical shell due to the of the restraints test environment. Figure 7 shows the test instrument and model. In experiment, the hammer was used to strike different positions of cylindrical shells in turn, and acceleration sensors with sensitivity of 100 mv/g were used to collect the vibration response at the same point. Then the time domain signals obtained by test were transformed into frequency domain signals by Fourier transform. The final results of frequencies are shown in Table 4. For natural frequencies obtained by FEM commercial program ABAQUS (S4R model), it is obvious that the structure and material parameters are the same as the experiment, and it should be note that the results have converge to stable when the mesh size is 0.03 m. From Table 4, it is easy to find that the present results closely agreed with experiment and FEM. For selected five modes, the maximum error of present method and experiment is 2.35%, and the maximum error of present method and FEM is 0.38%. The reason for the large error of present method with the test results are mainly the influence of elastic hoisting boundary and random error. The mode shapes obtained by three different methods are presented in Figure 8.

Table 5 exhibits the results of free vibration behaviors for stepped FG paraboloidal shell with various boundary conditions. From Table 5, it is easy to find that the free vibration characteristics are not only influence by boundary conditions, but material parameters. To better reveal the vibration characteristics of the shell, some mode shapes are given in Figure 9.

Table 6 shows the results of stepped FG paraboloidal shell with different power-law exponents, in which four values are included. From Table 6, we can get that the boundary conditions and power-law exponents all will have important impact on the results of the structure.

Table 7 shows the results of stepped FG paraboloidal shell with different thickness distributions. Four kinds of thickness distributions, i.e., *h*_1_:*h*_2_:*h*_3_:*h*_4_:*h*_5_ = 0.04:0.045:0.05:0.055:0.06 are included. It is obvious that the thickness distributions affect the vibration behavior of stepped FG paraboloidal shell largely.

Figure 10, Figure 11 and Figure 12 exhibit the frequency parameters Ω of stepped FG paraboloidal shell with various parameters *a*, *b*, *c* and *p*. From selected data, it could be found that *a*, *b* and *c* have a great deal of impact on the results of Ω. In addition, for parameters *a* and *c*, the smaller value will obtain the larger results. Figure 13 exhibits the results of stepped FG paraboloidal shell with various stiffness ratios and parameter *p*. It can be seen that no matter what value of parameter *p*, the vibration characteristics will decrease with *E_c_*/*E_m_* increasing.

## 4. Conclusions

The paper proposed a solving formulation to investigate the free vibration behaviors of stepped FG paraboloidal shell with general boundary conditions. The paper is based on multi-segment strategy and FSDT. The displacement functions are simulated by Jacobi polynomials and Fourier series. To obtain the general boundary conditions of stepped FG paraboloidal shell, the penalty method was adopted. The final modes solutions about FG paraboloidal shell were obtained by Rayleigh–Ritz method. The most discoveries of proposed method are unified Jacobi polynomials, which make the displacement functions easier to select. For convergence analysis, the influence of boundary parameters, numbers of shell segments etc. are examined. The accuracy of this method was verified by the comparison study with those obtained by published literature, FEM, and the experiment. The results of this paper can provide the reference data for future research.

## Figures and Tables

**Figure 1 materials-12-00069-f001:**
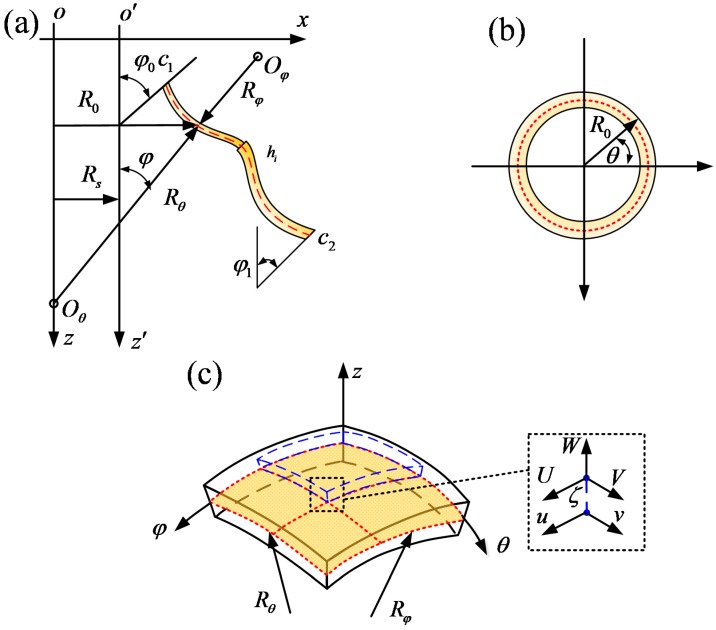
Geometry notations and coordinate system of stepped FG paraboloidal shell. (**a**) Geometric relationship; (**b**) cross-section; (**c**) coordinate system.

**Figure 2 materials-12-00069-f002:**
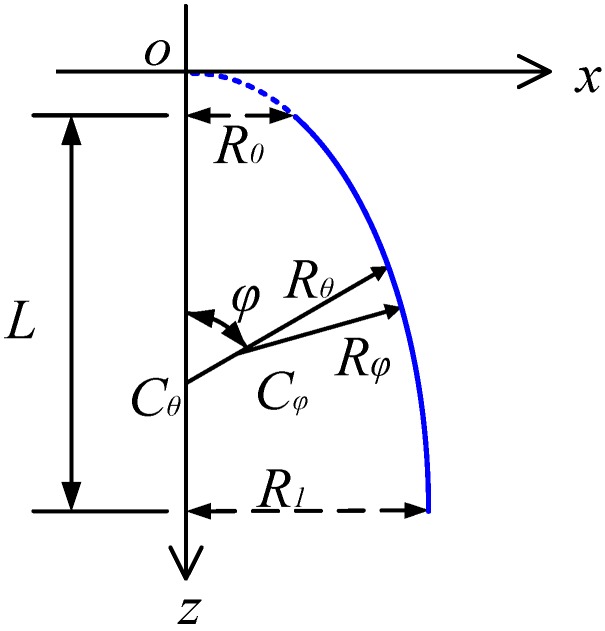
The geometric of doubly curved paraboloidal shell.

**Figure 3 materials-12-00069-f003:**
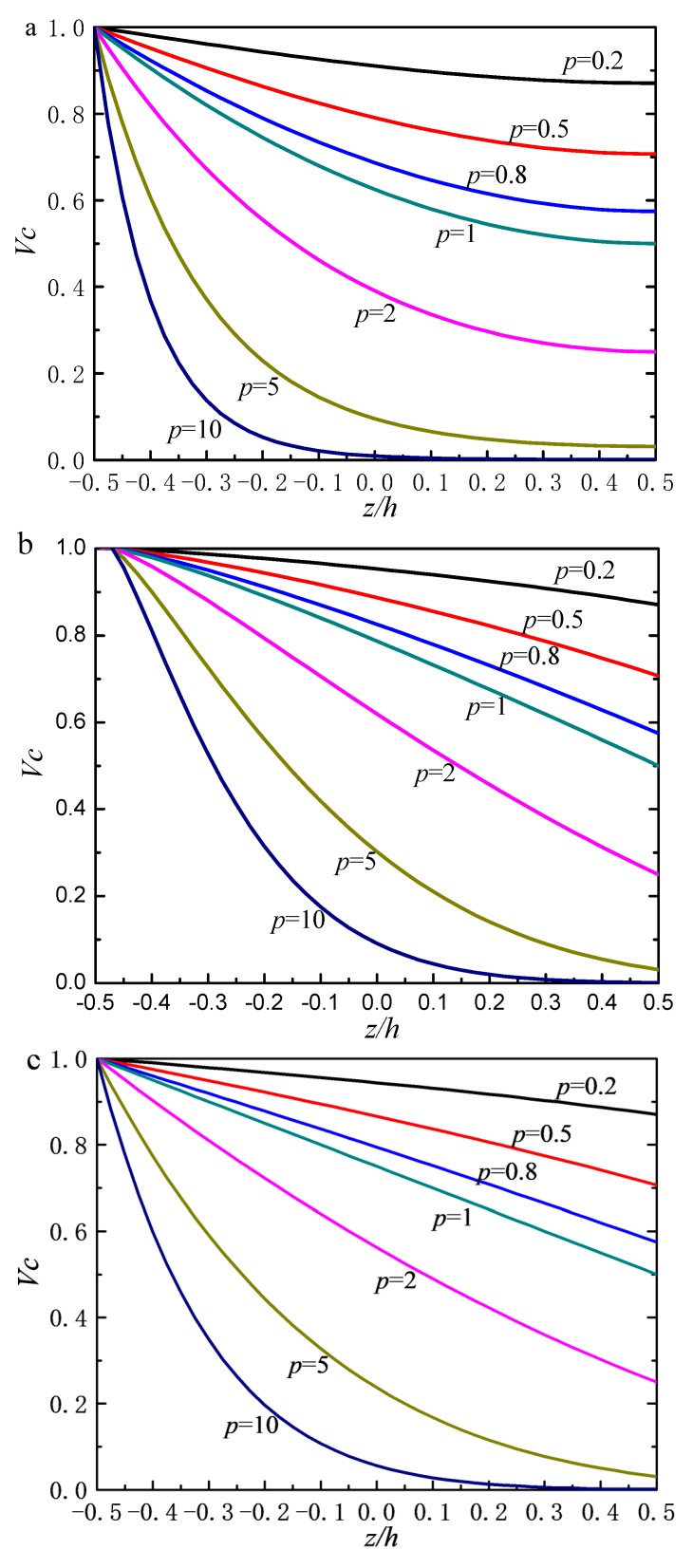
The variations *V_c_* about various values of *a*, *b*, *c* and *p*: (**a**) FGM_I_ (*a* = 0; *b* = 0.5; *c* = 2; *p*); (**b**) FGM_I_ (*a* = 1; *b* = 0.5; *c* = 0.8; *p*); (**c**) FGM_I_ (*a* = 0; *b* = −0.5; *c* = 1; *p*).

**Figure 4 materials-12-00069-f004:**
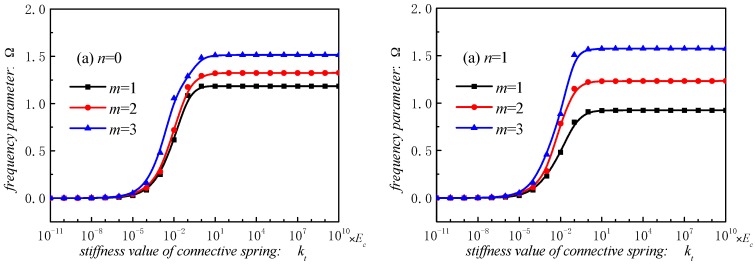
Frequency parameters Ω of stepped FG paraboloidal shell with various boundary parameters.

**Figure 5 materials-12-00069-f005:**
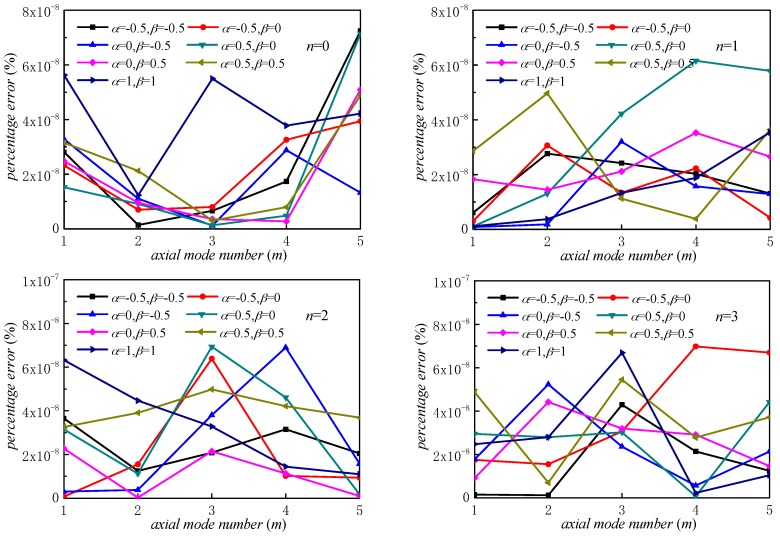
Relative error of frequency parameters Ω in stepped FG paraboloidal shell (BC: C–C).

**Figure 6 materials-12-00069-f006:**
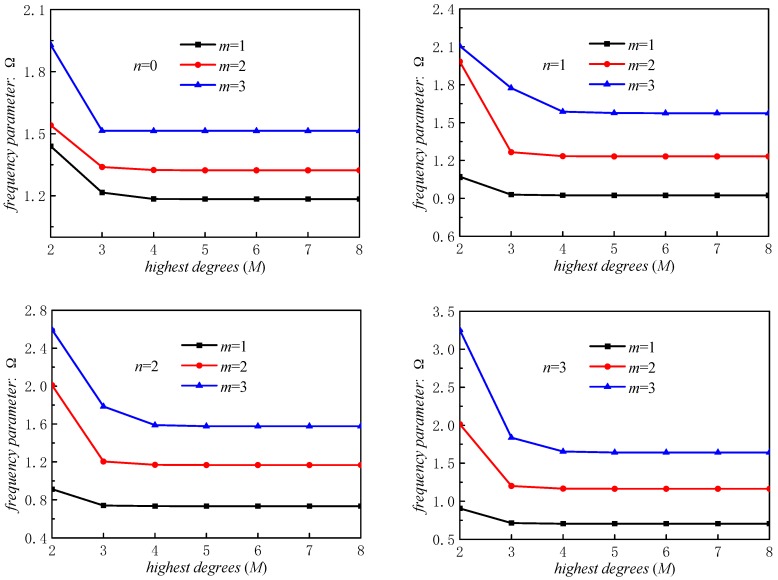
Frequency parameters Ω for various truncation in stepped FG paraboloidal shell.

**Figure 7 materials-12-00069-f007:**
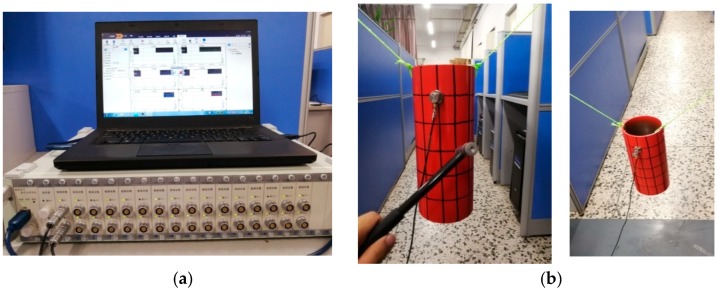
Testing instruments and model. (**a**) The test system; (**b**) the test model.

**Figure 8 materials-12-00069-f008:**
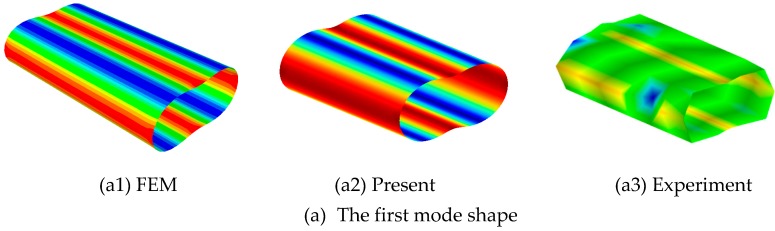
The selected mode shapes of three kinds of method.

**Figure 9 materials-12-00069-f009:**
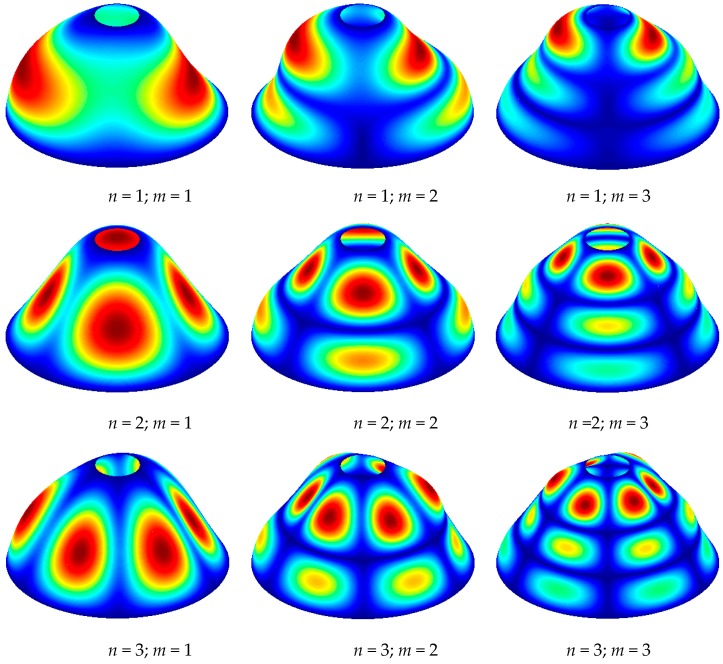
Mode shapes of stepped FG paraboloidal shell (BC: SS–SS).

**Figure 10 materials-12-00069-f010:**
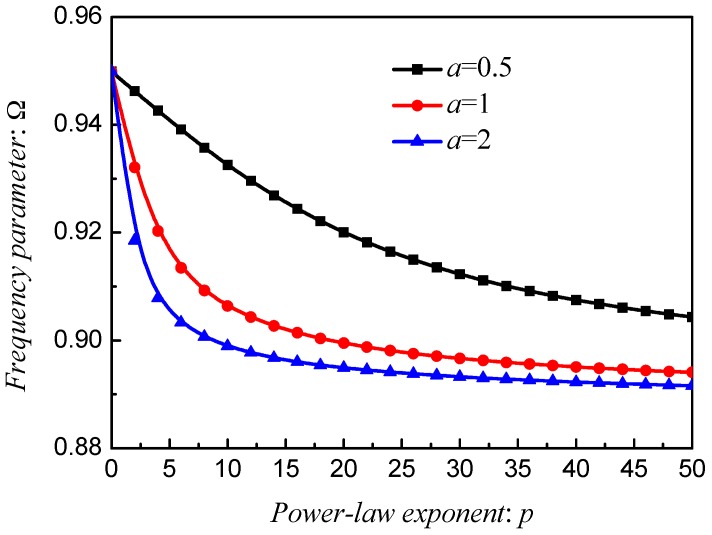
Results about different *p* and *a* of stepped FGM_I_ (*a*, *b* = 0.5; *c* = 2; *p*) paraboloidal shell.

**Figure 11 materials-12-00069-f011:**
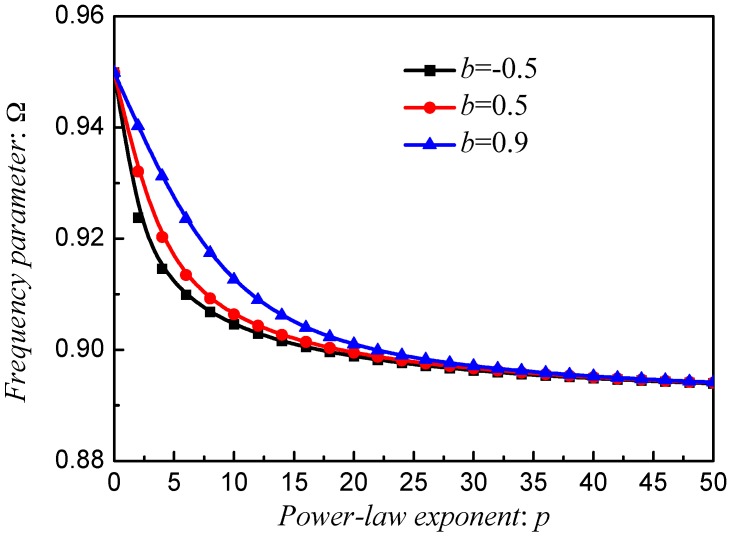
Results about different *p* and *b* of stepped FGM_I_ (*a* = 1; *b*, *c* = 2; *p*) paraboloidal shell.

**Figure 12 materials-12-00069-f012:**
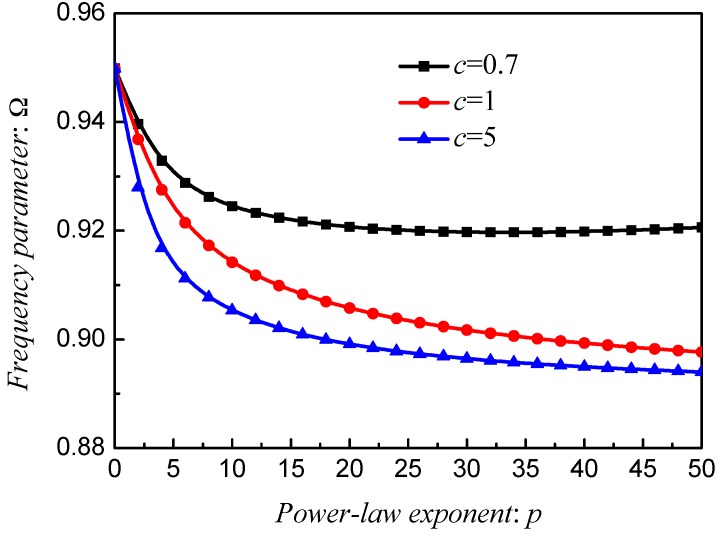
Results about different *p* and *c* of stepped FGM_I_ (*a* = 1; *b* = 0.5; *c*; *p*) paraboloidal shell.

**Figure 13 materials-12-00069-f013:**
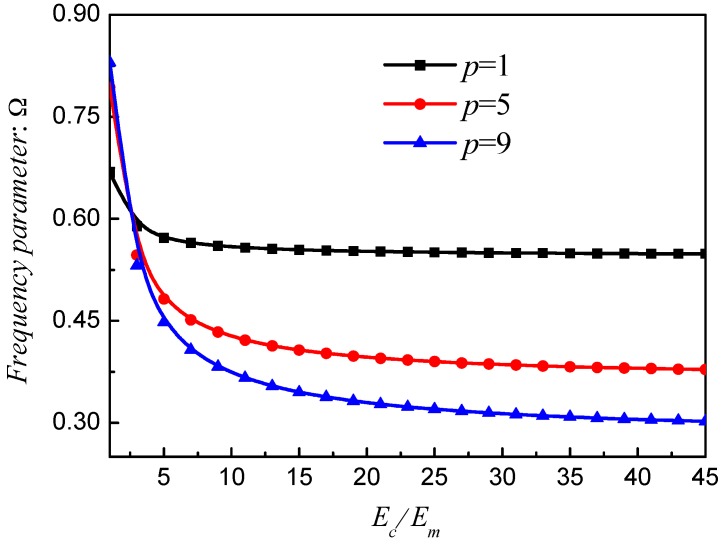
Results about different *E_c_*/*E_m_* and *p* of stepped FGM_I_ (*a* = 1; *b* = 0.5; *c* = 2; *p*) paraboloidal shell.

**Table 1 materials-12-00069-t001:** Spring stiffness values.

BC	*k_u_*_,0_, *k_u_*_,1_	*k_v_*_,0_, *k_v_*_,1_	*k_w_*_,0_, *k_w_*_,1_	*k_φ_*_,0_, *k_φ_*_,1_	*k_θ_*_,0_, *k_θ_*_,1_
F	0	0	0	0	0
SD	0	10^3^ *E_c_*	10^3^ *E_c_*	0	0
SS	10^3^ *E_c_*	10^3^ *E_c_*	10^3^ *E_c_*	0	10^3^ *E_c_*
C	10^3^ *E_c_*	10^3^ *E_c_*	10^3^ *E_c_*	10^3^ *E_c_*	10^3^ *E_c_*
E1	10^−3^ *E_c_*	10^3^ *E_c_*	10^3^ *E_c_*	10^3^ *E_c_*	10^3^ *E_c_*
E2	10^3^ *E_c_*	10^−3^ *E_c_*	10^3^ *E_c_*	10^3^ *E_c_*	10^3^ *E_c_*
E3	10^−3^ *E_c_*	10^−3^ *E_c_*	10^3^ *E_c_*	10^3^ *E_c_*	10^3^ *E_c_*

**Table 2 materials-12-00069-t002:** Frequency parameter Ω of the FGM_I_ (*a* = 1; *b* = 0; *c*; *p*) spherical shell structure (BC; C–C, *m* = 1).

Power-Law Exponent	Number of the Segment (*H_e_*)								Ref [46]
	*n*	2	3	4	5	6	7	8	
*p* = 0.6	1	1.0569	1.0569	1.0568	1.0568	1.0568	1.0568	1.0568	1.0538
	2	1.0379	1.0376	1.0374	1.0372	1.0371	1.0371	1.0370	1.0354
	3	1.0319	1.0317	1.0314	1.0312	1.0312	1.0310	1.0310	1.0294
	4	1.0760	1.0757	1.0755	1.0752	1.0751	1.0750	1.0749	1.0733
	5	1.1588	1.1586	1.1584	1.1581	1.1581	1.1580	1.1580	1.1559
*p* = 5	1	1.0446	1.0446	1.0446	1.0445	1.0445	1.0445	1.0445	1.0411
	2	1.0116	1.0115	1.0113	1.0111	1.0110	1.0109	1.0108	1.0085
	3	1.0085	1.0083	1.0082	1.0080	1.0079	1.0079	1.0078	1.0053
	4	1.0572	1.0571	1.0569	1.0568	1.0566	1.0565	1.0563	1.0539
	5	1.1470	1.1468	1.1467	1.1465	1.1464	1.1464	1.1463	1.1433
*p* = 20	1	1.0282	1.0282	1.0281	1.0281	1.0281	1.0281	1.0281	1.0266
	2	0.9958	0.9957	0.9956	0.9954	0.9953	0.9953	0.9952	0.9945
	3	0.9927	0.9926	0.9924	0.9923	0.9922	0.9921	0.9920	0.9913
	4	1.0407	1.0405	1.0404	1.0403	1.0403	1.0402	1.0399	1.0392
	5	1.1290	1.1289	1.1287	1.1286	1.1285	1.1284	1.1284	1.1273

**Table 3 materials-12-00069-t003:** Comparison of frequency parameter Ω for stepped doubly curved paraboloidal shell (FGM_I_ (*a*, *b*, *c*, *p* = 0)).

*n*	*m*	Proposed Method	FEM
0	1	1.2139	1.2144
	2	1.3579	1.3586
	3	1.5621	1.5645
	4	1.6154	1.6183
1	1	0.9499	0.9504
	2	1.2605	1.2615
	3	1.6030	1.6070
	4	1.9770	1.9725
2	1	0.7521	0.7524
	2	1.1907	1.1924
	3	1.6002	1.6056
	4	2.1071	2.1083
3	1	0.7171	0.7176
	2	1.1811	1.1835
	3	1.6590	1.6566
	4	2.2217	2.2251

**Table 4 materials-12-00069-t004:** Comparison study of the frequencies for cylindrical shell.

*n*, *m*	Present	Experimental	Error (%)	FEM	Error (%)
0, 1	545.89	551.97	1.11	547.49	0.29
2, 2	582.13	588.39	1.08	581.98	0.03
0, 3	1561.93	1572.53	0.68	1567.90	0.38
2, 3	1618.37	1656.42	2.35	1613.70	0.29
3, 3	2143.98	2169.05	1.17	2150.70	0.31

**Table 5 materials-12-00069-t005:** Frequency parameters Ω of stepped paraboloidal shell.

Type	*n*	*m*	Boundary Restraints									
			F–C	C–C	SD–SD	SS–SS	E1–E1	E2–E2	E3–E3	F–E1	F–E2	F–SS
FGM_I_ (*a* = 1; *b* = −0.5; *c* = 2; *p* = 2)	1	1	0.7470	0.9238	0.6151	0.8886	0.6736	0.5307	0.2076	0.2171	0.4519	0.7301
		2	1.1767	1.2318	0.8874	1.1522	0.9171	1.2104	0.5468	0.8334	1.1066	1.0967
		3	1.4199	1.5746	1.1503	1.4582	1.2343	1.4709	1.1710	1.1866	1.4113	1.3661
		4	1.6262	1.9378	1.4683	1.8208	1.5746	1.7016	1.4638	1.4535	1.4271	1.6000
		5	1.7717	2.0975	1.8376	2.0156	2.0259	1.9062	1.5645	1.7696	1.7209	1.6947
	2	1	0.5453	0.7334	0.6855	0.6973	0.7196	0.6154	0.5819	0.4742	0.4983	0.5329
		2	0.9432	1.1666	1.0460	1.0800	1.1426	1.1078	0.9250	0.9212	0.8446	0.8894
		3	1.3195	1.5767	1.3190	1.4539	1.3368	1.5389	1.3106	1.3023	1.3013	1.2305
		4	1.7651	2.0848	1.4434	1.9063	1.5690	2.0412	1.4790	1.3588	1.7237	1.6529
		5	2.2154	2.5207	1.8933	2.4718	2.0823	2.2287	1.9788	1.7737	2.1426	2.1436
	3	1	0.6918	0.7037	0.6469	0.6597	0.6939	0.6630	0.6588	0.6816	0.6549	0.6516
		2	1.1043	1.1629	1.0674	1.0763	1.1594	1.1255	1.0998	1.1029	1.0725	1.0362
		3	1.4964	1.6405	1.5063	1.5173	1.6362	1.6151	1.5855	1.4963	1.4798	1.4098
		4	1.9638	2.2032	1.9827	2.0370	2.0021	2.1782	1.8904	1.9536	1.9463	1.8532
		5	2.5376	2.8897	2.0261	2.6731	2.1986	2.8331	2.1627	2.0105	2.5267	2.4127
FGM_II_ (*a* = 1; *b* = −0.5; *c* = 2; *p* = 2)	1	1	0.7418	0.9171	0.6086	0.8875	0.6689	0.5274	0.2064	0.2161	0.4491	0.7153
		2	1.1663	1.2205	0.8821	1.1388	0.9110	1.1991	0.5427	0.8263	1.0974	1.1007
		3	1.4059	1.5589	1.1408	1.4569	1.2232	1.4590	1.1604	1.1759	1.4010	1.3328
		4	1.6117	1.9209	1.4556	1.7561	1.5588	1.6896	1.4532	1.4381	1.4141	1.6112
		5	1.7523	2.0820	1.8189	2.0554	2.0045	1.8919	1.5520	1.7494	1.7022	1.6794
	2	1	0.5378	0.7271	0.6821	0.6895	0.7136	0.6103	0.5768	0.4670	0.4920	0.5171
		2	0.9331	1.1553	1.0399	1.0802	1.1315	1.0968	0.9159	0.9117	0.8351	0.8909
		3	1.3062	1.5593	1.3101	1.4468	1.3285	1.5218	1.3017	1.2916	1.2878	1.2172
		4	1.7468	2.0615	1.4329	1.8881	1.5518	2.0196	1.4630	1.3468	1.7066	1.6445
		5	2.1931	2.5059	1.8788	2.4820	2.0593	2.2170	1.9600	1.7554	2.1281	2.0916
	3	1	0.6846	0.6968	0.6436	0.6477	0.6872	0.6566	0.6524	0.6746	0.6482	0.6389
		2	1.0906	1.1506	1.0609	1.0731	1.1471	1.1136	1.0877	1.0892	1.0592	1.0300
		3	1.4762	1.6214	1.4944	1.5073	1.6171	1.5961	1.5665	1.4761	1.4597	1.3933
		4	1.9404	2.1772	1.9727	2.0216	1.9913	2.1526	1.8796	1.9316	1.9231	1.8339
		5	2.5057	2.8601	2.0097	2.6553	2.1728	2.8113	2.1386	1.9983	2.4952	2.3801

**Table 6 materials-12-00069-t006:** Frequency parameters Ω for stepped FGM_I_ (*a* = 1; *b* = 0.5; *c* = 2; *p*) shell with different power-law exponents.

Power-Law Exponents	*n*	*m*	C–C	SD–SD	F–SS
*p* = 0.2	1	1	0.9480	0.6315	0.7461
		2	1.2583	0.9133	1.1301
		3	1.6007	1.1769	1.3859
	2	1	0.7507	0.7047	0.5374
		2	1.1888	1.0703	0.9119
		3	1.5983	1.3557	1.2499
	3	1	0.7161	0.6612	0.6606
		2	1.1796	1.0868	1.0534
		3	1.6574	1.5264	1.4231
*p* = 0.5	1	1	0.9451	0.6297	0.7444
		2	1.2550	0.9104	1.1259
		3	1.5971	1.1737	1.3836
	2	1	0.7486	0.7025	0.5369
		2	1.1859	1.0673	0.9090
		3	1.5951	1.3514	1.2472
	3	1	0.7144	0.6593	0.6595
		2	1.1772	1.0841	1.0508
		3	1.6545	1.5232	1.4207
*p* = 2	1	1	0.9321	0.6210	0.7359
		2	1.2395	0.8969	1.1078
		3	1.5796	1.1585	1.3711
	2	1	0.7389	0.6922	0.5332
		2	1.1721	1.0532	0.8959
		3	1.5789	1.3322	1.2339
	3	1	0.7064	0.6508	0.6534
		2	1.1650	1.0712	1.0389
		3	1.6394	1.5074	1.4081
*p* = 5	1	1	0.9164	0.6103	0.7236
		2	1.2221	0.8807	1.0892
		3	1.5625	1.1413	1.3542
	2	1	0.7276	0.6803	0.5280
		2	1.1575	1.0383	0.8832
		3	1.5646	1.3084	1.2210
	3	1	0.6983	0.6422	0.6460
		2	1.1540	1.0597	1.0285
		3	1.6282	1.4954	1.3991

**Table 7 materials-12-00069-t007:** Frequency parameters Ω for stepped FGM_I_ (*a* = 1; *b* = 0.5; *c* = 2; *p* = 2) shell with different thickness distributions.

*h*_1_:*h*_2_:*h*_3_:*h*_4_:*h*_5_	*n*	*m*	C–C	SD–SD	F–SS
0.04:0.05:0.06:0.07:0.08	1	1	0.9579	0.5884	0.7655
		2	1.3085	0.9008	1.1470
		3	1.6903	1.2140	1.4461
	2	1	0.7667	0.6952	0.5476
		2	1.2454	1.1009	0.9267
		3	1.7064	1.2969	1.3145
	3	1	0.7590	0.6841	0.6925
		2	1.2600	1.1513	1.1067
		3	1.7917	1.6410	1.5140
0.08:0.07:0.06:0.05:0.04	1	1	0.8600	0.6915	0.6176
		2	1.1979	0.9477	1.0841
		3	1.6283	1.1045	1.2400
	2	1	0.7026	0.6680	0.5916
		2	1.1782	1.0529	0.9674
		3	1.6697	1.4977	1.3075
	3	1	0.6992	0.6584	0.6578
		2	1.2297	1.1176	1.1192
		3	1.8009	1.6410	1.6103
0.04:0.06:0.08:0.07:0.05	1	1	0.8483	0.5982	0.6988
		2	1.2766	0.8162	1.1448
		3	1.6965	1.1861	1.3817
	2	1	0.6747	0.6343	0.5059
		2	1.2266	1.0603	0.9036
		3	1.7104	1.4039	1.2943
	3	1	0.6993	0.6493	0.6457
		2	1.2530	1.1403	1.1109
		3	1.8046	1.6523	1.5196
0.07:0.05:0.04:0.06:0.08	1	1	1.0086	0.6664	0.7052
		2	1.2356	0.9912	1.0789
		3	1.6421	1.1382	1.4098
	2	1	0.8278	0.7647	0.6457
		2	1.1948	1.0710	0.9595
		3	1.6748	1.2893	1.3483
	3	1	0.8000	0.7185	0.7324
		2	1.2351	1.1212	1.1102
		3	1.7878	1.6336	1.5970

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
