# Peer review of "Application of First-Order Shear Deformation Theory on Vibration Analysis of Stepped Functionally Graded Paraboloidal Shell with General Edge Constraints"

_materials, 2018, doi:10.3390/ma12010069_

Reviewer 1 Report

Please follow editorial and crtitical remarks introduced into the manuscript.

Author Response

The point-to-point answers and explanations for all comments have been made and listed in the following attachment.

Reviewer 2 Report

The paper aims to present a new method to analyze the vibration characteristic of stepped functionally graded (FG) paraboloidal shell structures with general edge restraints. In the reviewer’s opinion, the topic is interesting and discussed in a complete manner. A minor revision is required to improve the following aspects:

1. The use of English could be improved noticeably throughout the whole manuscript.

2. Further comments could be added to illustrate in a more complete manner the differences between the two expression of Vc in equations (2).

3. The graphs in figure 5 could be more discussed.

4. The experimental test could be described by adding further details and comments.

5. Analogously, the FEM model which has provided the results in table 4 should be discussed more (number of elements, degrees of freedom, commercial software,…).

Author Response

(The authors gave the same response as above.)

Reviewer 3 Report

The authors investigate the vibration response of stepped functionally graded paraboloidal shells with general edge restraints. The study is quite comprehensive, including experiments, theory and simulations. However, a number of major issues have been identified:

1) The novelty of the work is not clear. The papers referred to in the introduction deal with general methods that can be applicable to the specific case of a stepped FG paraboloidal shell structure analysed here. Where is then the novelty of the method? Why is it better than other approaches?

2) The authors have published a number of very similar studies, and this work appears to be a case study that could be a section of one of their previous papers. I am referring to Composites Part B: Engineering 164, pp. 249-264; European Journal of Mechanics, A/Solids 74, pp. 48-65; Composite Structures 201, pp. 86-111, International Journal of Mechanical Sciences 145, pp. 64-82 and references therein. The authors should clarify in their response (not necessarily on the paper) why the present work is of sufficient entity to constitute a paper of its own.

3) The finite element methodology employed is not clear. The authors need to clearly specify how they are introducing the material gradient. Are they using homogeneous elements or graded elements? This discussion needs to be done in the context of existing literature (“Numerical analysis of quasi-static fracture in functionally graded materials” IJMMD 2015 is probably the most relevant reference in this regard).

Author Response

The point-to-point answers and explanations for all comments have been made and listed in the following attachment.

Round  2

Reviewer 3 Report

The authors have very diligently addressed my concerns. The novelty of the paper has now been made clear. While there isn't a big piece of novelty, there are several incremental contributions to the existing literature and the paper is, overall, a nice and extensive piece of work, combining experiments, theory and modeling. Publication is now recommended.